# The Inhibitory Effect of Magnolol on the Human TWIK1 Channel Is Related to G229 and T225 Sites

**DOI:** 10.3390/molecules28196815

**Published:** 2023-09-27

**Authors:** Jintao Wang, Huan Liu, Zhuolin Sun, Xinyi Zou, Zixuan Zhang, Xiaofeng Wei, Lanying Pan, Antony Stalin, Wei Zhao, Yuan Chen

**Affiliations:** 1Zhejiang Provincial Key Laboratory of Resources Protection and Innovation of Traditional Chinese Medicine, College of Food and Health, Zhejiang Agriculture and Forestry University, Hangzhou 311300, China; 2020602052011@stu.zafu.edu.cn (J.W.); 2020602052004@stu.zafu.edu.cn (H.L.); 18275995437@163.com (Z.S.); zouxinyi00@163.com (X.Z.); 2021102062003@stu.zafu.edu.cn (Z.Z.); 2020102131011@stu.zafu.edu.cn (X.W.); weizhao@zafu.edu.cn (W.Z.); 2Shulan International Medical College, Zhejiang Shuren University, Hangzhou 310015, China; 3Institute of Fundamental and Frontier Sciences, University of Electronic Science and Technology of China, Chengdu 610054, China; antonystalin@uestc.edu.cn

**Keywords:** magnolol, two-pore domain potassium channel, TWIK1

## Abstract

TWIK1 (K2P1.1/KCNK1) belongs to the potassium channels of the two-pore domain. Its current is very small and difficult to measure. In this work, we used a 100 mM NH_4_^+^ extracellular solution to increase TWIK1 current in its stable cell line expressed in HEK293. Then, the inhibition of magnolol on TWIK1 was observed via a whole-cell patch clamp experiment, and it was found that magnolol had a significant inhibitory effect on TWIK1 (IC_50_ = 6.21 ± 0.13 μM). By molecular docking and alanine scanning mutagenesis, the IC_50_ of TWIK1 mutants G229A, T225A, I140A, L223A, and S224A was 20.77 ± 3.20, 21.81 ± 7.93, 10.22 ± 1.07, 9.55 ± 1.62, and 7.43 ± 3.20 μM, respectively. Thus, we conclude that the inhibition of the TWIK1 channel by magnolol is related to G229 and T225 on the P2- pore helix.

## 1. Introduction

*Magnolia officinalis* is a Chinese herbal medicine. It is listed in “Shennong’s Herbal” and “Compendium of Materia Medica” and is used as a medicine with dry bark, root bark, and branch bark. In traditional medicine, *M. officinalis* plays many roles, including relieving asthma, eliminating phlegm, and relieving pain. In clinical application, its compatibility with various Chinese medicines is widely used to treat cough, asthma, abdominal distension, constipation, malaria, and other diseases [1].

Magnolol is an effective main ingredient of *M. officinalis*. Studies have shown that magnolol possesses a wide range of pharmacological activities in vitro and in vivo. It can interfere with TLR4/NF-κB/MAPK, PPAR-γ/NF-κB and activate the PI3K/Akt/NF-κB signaling pathway, decrease the expression of inflammatory cytokines, and exhibit anti-inflammatory activity [1,2,3,4]. By attenuating the AKT/GSK3β/β-catenin/ERK1/2 and HIF-1α/VEGF signaling pathways, it inhibited vascular smooth muscle cell proliferation and migration, providing cardiovascular protection [1,5,6]. It exerts a neuroprotective effect by inhibiting the production of PGE2 [1,7].

In addition, it can enhance adipocyte differentiation and glucose uptake by dual activation of RXRα/PPAR-γ and inhibition of α-glucosidase, suggesting hypoglycemic activity [1,8,9]. Therefore, the presence of glucose has been reported to depolarize TWIK1 in pancreatic β-cells [10]. This brings us to the question of whether magnolol can regulate blood glucose by directly affecting the TWIK1 channel.

The two-pore domain potassium channel (K2P) is a background potassium channel encoded by 15 KCNK genes. According to its sequence homology, it can be divided into six subfamilies: TWIK, TREK, TASK, THIK, TALK, and TRESK [11,12,13,14,15,16]. These channels have a similar structure. All of them have two subunits coupled into a dimer consisting of two pore domains (P1 and P2), four transmembrane helices (M1–M4), and a large extracellular M1P1 loop. TWIK1 was discovered in 1996 in the human kidney cDNA library and is significantly expressed in the heart, brain, pancreas, and other tissues. It mediates glutamate release and ammonia uptake in the central and peripheral nervous systems [17,18,19]. The current of wild-type TWIK1 is very small and difficult to measure and characterize. Therefore, it is urgent to develop a method to directly and effectively measure TWIK1 current.

Regarding the unique “silencing” mechanism of TWIK1, it was initially hypothesized that TWIK1 is associated with small ubiquitin-related modifier proteins (SUMO) due to an unconventional site (L274) in TWIK1, resulting in channel silencing, and that mutation of this site to glutamic acid (K274E) could activate TWIK1 current [20]. However, it was soon found that the increase in TWIK1 current density caused by the K274E mutation had nothing to do with SUMO inhibition [20,21,22]. However, although the main mechanism of “silencing” was investigated, it became the main reason for the increase in TWIK1 current by mutation of certain amino acid residues [10,20,23]. Recently, it was reported that the silent TWIK1 channel in CHO cells transfected with TWIK1 can conduct monovalent cation currents (K^+^, NH_4_^+^) when extracellular ionic conditions change, and it was found that the inward NH_4_^+^ current of the TWIK1 channel was 40 times larger than the K^+^ current in a heterologous expression system [24].

In this study, we replaced the same Na^+^ concentration with high NH_4_^+^ concentrations to activate the TWIK1 current in HEK293-TWIK1 cells for the subsequent experiments. Then, the concentration-dependent inhibition of magnolol on TWIK1 was observed by a whole-cell patch clamp experiment, subsequently the actual binding site of magnolol and TWIK1 was determined by molecular docking prediction. The mutation was used to investigate whether the effect of magnolol on TWIK1 was related to the binding site between amino acid residues in the selective filter.

## 2. Results

### 2.1. TWIK1 Is Activated by High K^+^ and High NH_4_^+^ Extracellular Solution

We used 10, 30, 100 mm NH_4_Cl or KCl to replace the same concentration of NaCl to activate the current of stably transfected TWIK1 cells. The results show that as the ion concentration increases, the difference between the activation efficiencies of the two different ions becomes more significant, and the maximal peak current is at 100 mM in the NH_4_^+^ extracellular solution (Figure 1). Therefore, we used the 100 mM NH_4_^+^ extracellular fluid for our screening experiments.

### 2.2. Magnolol Significantly Inhibited the Activation Current of TWIK1

We perfused TWIK1 cells with magnolol using the whole-cell patch clamp technique and found that magnolol inhibited TWIK1 at low concentrations. Further investigation revealed that the inhibition was concentration-dependent, as we observed that TWIK1 current decreased with increasing magnolol concentration. The IC50 is 6.21 ± 0.13 μM (Figure 2b).

### 2.3. Molecular Docking of Magnolol with TWIK1

We predicted the interaction of magnolol with TWIK1 by molecular docking using the AutodockTools software. When the docking box is placed on the active ligand position of the TWIK1 protein, five optimal docking results were obtained. The predicted docking results suggest that there can be hydrogen bonds between magnolol and G229 at the helix of the P2 hole when the docking box covers the entire TWIK1 channel structure (Figure 3a). When the docking box contains the P2 pore domain of TWIK1, magnolol can form hydrogen bonds with L223 on the P2 pore helix (Figure 3b). However, when the docking box contains the P1 and P2 pore domains of TWIK1, magnolol can form hydrogen bonds with S224 and T225 on the P2 pore helix, M260 on the M4 transmembrane helix, and I140 on the M2 transmembrane helix (Figure 3b–d). This suggests that magnolol interacts with two pore domains (P1 and P2) and transmembrane helices (M2 and M4) that channel the gating of TWIK1, resulting in inhibition.

### 2.4. The Comparison of the Inhibitory Effect of Magnolol on Wild-Type and Different Mutation TWIK1

The “alanine scanning” method experimentally verifies the results of the molecular docking prediction. The IC_50_ of magnolol for the mutant of T225A is 21.81 ± 7.93 μM and that of G229A is 20.77 ± 3.20 μM, which is significantly higher than the IC_50_ of 6.21 ± 0.13 μM of the wild-type channel (Figure 4b–d; *n* = 5). Therefore, the data suggest that G229 and T225 are important amino acid residues at which magnolol binds to the TWIK1 channel. The IC_50_s of the I140A and L223A mutations are slightly higher than those of the wild type: 10.22 ± 1.07 μM and 9.55 ± 1.62 μM, respectively (Figure 4e,f; *n* = 5). Therefore, we speculate that these two sites also play a role in the blocking of the TWIK1 activation current by magnolol. The IC_50_ of the last S224A mutation is almost the same as that of the wild type: 7.43 ± 3.20 μM (Figure 4g; *n* = 5). It is unlikely that this site plays a role in the binding of magnolol to TWIK1 alone. Taken together, our data suggest that the inhibitory effect of magnolol on TWIK1 may be related to the hydrogen bonding of G229, T225, and L223 at the P2 pore helix and I140 at the M2 transmembrane helix.

## 3. Discussion

In the regular expression system, the current amplitude of TWIK1 is tiny and difficult to measure. This is the main reason why it is difficult to study the electrophysiological characteristics and functions of the channel. We activated TWIK1 by replacing the same Na^+^ concentration (to maintain Osmolarity) with different concentrations of high K^+^ or NH_4_^+^ extracellular solutions to obtain a much larger current. This is related to the exceptional dynamic ion-selective properties of TWIK1 [10,25,26,27]. The TWIK1 channel is permeable to Na^+^ in an acidic pH or low K^+^ environment, and this permeability to Na^+^ is considered unique among potassium channels, which is also the main reason for the abnormal depolarization of human cardiomyopathy in hypoglycemia [26,27]. In astrocytes, it is an efficient NH_4_^+^ uptake channel that maintains the acidic internal environment of the cells. This channel can be regulated in vivo by metabotropic glutamate receptor 3 (mGluR3), and activation of mGluR3 can promote the recruitment of TWIK1 to the astrocyte membrane and enhance NH_4_^+^ uptake [19]. Therefore, we used a 100 mM NH_4_^+^ extracellular solution for the subsequent cell perfusion experiments.

Many studies appear that the combination of K2P channels and drugs mainly occurs at the inner hole or inner membrane fenestration under the selective filter, blocking the ion conduction pathway in the intracellular cavity [28,29]. Our experiment suggests that magnolol, the main ingredient of *M. officinalis*, enters the TWIK1 channel and binds in the inner hole of the selective filter membrane, resulting in a concentration-dependent inhibition of the TWIK1 channel, with the IC_50_ reaching 6.21 μM. We then applied the molecular docking method and the alanine scanning mutagenesis experiment to observe with which sites on the pore helix and transmembrane helix of the TWIK1 channel magnolol might interact.

According to the results of molecular docking prediction, magnolol interacts with amino acid residues (G229, T225, S224, L223, I140, M260) around the pores of the selective filter to form hydrogen bonds when the TWIK1 channel is open. These hydrogen bonds could inhibit the TWIK1 current. The selective filter of the TWIK1 channel is a typical conducting conformation with four major K^+^ binding sites (S1–S4) [30]. In the P2 domain, the selective filter (T226, G227, G229), loops (D230, Y231), and pore helices (Y217, I221) form a variable hydrogen bonding network together with water molecules [30]. When magnolol forms hydrogen bonds with G229 in the selective filter, it alters the interaction of amino acid residues in the P2 domain and prevents the binding of K^+^ at the S0–S2 site of the selective filter [30]. The T225 residue is the major lining residue of the intra-subunit fenestrations within the TWIK1 channel [31]. Its carbonyl group replaces the hydrated water of K^+^ to form hydrogen bonds with magnolol, and mutation of T225 can block K^+^-dependent inhibition and have a profound impact on the gating mechanism of the selective filter [32]. Our experimental results are consistent with this report. The T225 and G229 mutations significantly reduce the inhibitory effect of magnolol on TWIK1. Therefore, residues G229 and T225 are important amino acid sites for magnolol to inhibit TWIK1.

In a study, it was found that a construction site is observed in the fenestration radius from the outside to the inside of the membrane toward the central hole. The shrinkage of this position is associated with the presence of residues V139, I142, P143, F220, S224, M260, and L264 in TWIK1 [31]. However, we found that the S224 residue is not sensitive to alanine substitution and its IC_50_ differs little from that of the wild type, while no current flows after mutation of M260. Therefore, we thought that S224 and M260 were not important amino acid sites for blocking the TWIK1 channel by magnolol.

Finally, mutation of L223 and I140 also reduced the inhibitory effect of magnolol on TWIK1 but had little effect, so we also thought that I140 and L223 might not be the important amino acid sites for inhibition of TWIK1 by magnolol.

In this work, the mechanism of magnolol inhibiting the TWIK1 channel was studied. Furthermore, T225 and G229 residues were identified by mutation site as important amino acid residues that affected magnolol’s inhibition of the TWIK1 channel. This provides a research direction for us to discuss whether magnolol can affect the treatment of diabetes through specific amino acid residues, and also provides a new way for screening candidate drugs targeting the TWIK1 channel.

Thus, we conclude that the inhibition of the TWIK1 channel by magnolol is related to G229 and T225 on the P2 pore helix.

## 4. Materials and Methods

### 4.1. Materials

The cDNAs for the human TWIK1 channels (NM_002245.4) was bought from OriGene Technologies (Rockville, MD, USA), subcloned into pcDNA3.1 (neo+), and stably transfected into human embryonic kidney (HEK293) 293 cells, which was used to determine IC_50_s of magnolol in TWIK1 channel. Magnolol (HPLC ≥ 98%) was purchased from Shanghai Yuanye Biotechnology Co., Ltd. (Shanghai, Yuanye, China). NaCl, KCl, HEPES, D-(+)-Glucose, KASP, EGTA, CaCl_2_, MgCl_2_, and DMSO were purchased from Sigma-Aldrich (Louis, MO, USA).

### 4.2. T225A, G229A, L223A, S224A and I140A Mutations

Mutation PCR was performed with primers synthesized by Sangon biotech (Shanghai, China) using the KOD DNA polymerase (TOYOBO, Osaka, Japan). PCR products were digested by the DpnI restriction enzyme (TAKARA, Dalian, China), purified, and transformed into DH5α *E. coli* competent cells. After sequencing, plasmids with the desired mutations were extracted.

### 4.3. Cell-Line Preparation

TWIK1 stable cell line and HEK293 cells were cultured in a solution containing 10% (*v*/*v*) fetal bovine serum (FBS), 1% (*v*/*v*) P/S (100U Penicillin and 0.1 mg/mL streptomycin). An additional 100 μg/mL G418 were added to culture TWIK1 stable cell line. The cells were split, washed twice with a pipette gun, digested with 180 μL trypsin, cultured with 10% (*v*/*v*) fetal bovine serum (FBS), 1% (*v*/*v*) P/S (100U penicillin, and 0.1 mg/mL streptomycin). Transient transfection was performed on naïve HEK293 cells, which were transfected when they reached 40–70% confluence. A transient transfection reagent (Qiagen, Hilden, Germany) was used for transient transfection with 1 μg WT and T225A, G229A, L223A, S224A, and I140A mutations. CD8 plasmid was co-transfected. The successfully transfected cells were identified by labeling with CD8-specific antibody-coated microspheres (Dynal, Oslo, Norway).

### 4.4. Magnolol Preparation

The magnolol used in the experiment was accurately weighed according to its molecular weight. Then, it was dissolved in DMSO to prepare a 3 mM high-concentration stock solution, which was stored at −80 °C. On the day of the whole-cell patch-clamp experiment, the high-concentration stock solution was added to DMSO to prepare 9, 3, 0.9, 0.3, and 0.09 mM stock solution, respectively. Finally, 100 μL of the corresponding stock solution was added to 30 mL of the extracellular solution to prepare the medicinal liquid with a concentration gradient of 30, 10, 3, 1, and 0.3 μM for practical use.

### 4.5. Electrophysiology

The stable expression of TWIK1 were cells cultured for 12–24 h after split and then whole-cell patch clamp recordings were performed. A PC505B amplifier (Warner Instrument Corporation) was used to voltage-clamp isolated cells. The electrodes had a resistance of 3–5 MΩ. We applied a voltage protocol that stimulated TWIK1 channels from −20 mV to −100 mV for 2 s, and then returned to −20 mV. Clampex software (Version 10.4, Axon Instruments) was used to generate the protocol. The recordings were carried out at room temperature (25 ± 1 °C) [33]. Clampfit (Molecular Devices, San Jose, CA, USA) was used to obtain the current traces and to analyze data. The intracellular solution contained 10 mM HEPES, 130 mM KASP, 5 mM EGTA, 5 mM MgCl_2_, pH 7.4. The extracellular solution contained 37 mM NaCl, 100 mM NH4Cl, 4 mM KCl, 10 mM HEPES, 10 mM Glucose, 1 mM MgCl_2_, 1.8 mM CaCl_2_, pH 7.4.

### 4.6. Molecular Docking

The Autodock Vina software (http://vina.scripps.edu/) (accessed on 17 September 2021) prepared the ligands and proteins needed for molecular docking analysis. For the target protein, the crystal structure of TWIK1 (PDB ID: 3UKM) was obtained from the PDB database (https://www.rcsb.org/) (accessed on 18 September 2021) and the pretreatment with Autodock software included removing water molecules, adding hydrogen atoms, amino acid modifications, energy optimization, and force field parameter adjustment. The compound magnolol downloaded from the PubChem database (https://pubchem.ncbi.nlm.nih.gov/) (accessed on 20 September 2021) satisfied the low-energy conformation of the ligand structure. Vina in Pyrx software (https://pyrx.sourceforge.io/) (accessed on 20 September 2021) was used to perform molecular docking between the target protein and the active compound magnolol, and the results were then visualized using PyMOL software.

### 4.7. Data Analysis

All data were expressed as mean ± SEM (*n* = cell number). Electrophysiological data were analyzed using Clampfit software (Version 10.4, Axon Instruments, Molecular Devices, San Jose, CA, USA) and Origin 8.0 (OriginLab Corporation, Electronic Arts Inc, Redwood City, CA, USA). The IC_50_ value was determined by fitting the concentration-dependent data to the following Hill equation: I (%) = 1/{1 + (IC_50_/[D])^n^}. In the equation, I% represents the percentage inhibition of current amplitudes, IC_50_ represents the concentration of half maximal inhibition, [D] represents the concentration of a compound, and n represents the Hill coefficient. Graphpad Prism 9.5 was also used for data statistics.

## Figures and Tables

**Figure 1 molecules-28-06815-f001:**
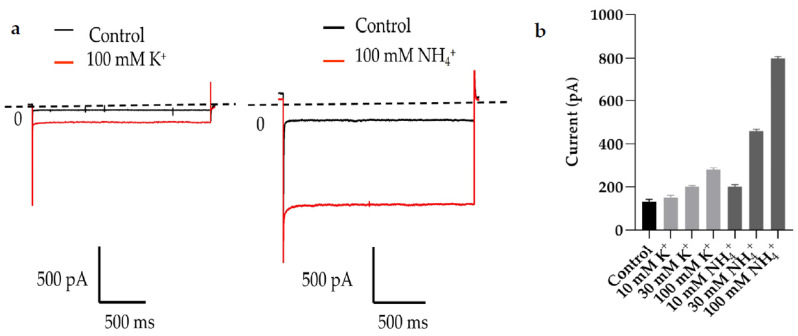
Current amplitude of TWIK1 at different K^+^ and NH_4_^+^ concentrations. (**a**) Left panel shows the example current traces at 100 mM K^+^. The right panel shows the example current traces at 100 mM NH4^+^. Using a standard protocol, we activated the TWIK1 currents by evoking the cells with a 2 s test pulse of 100 mV from a holding potential of −20 mV. Black is the current trace before activation and red is the current trace after activation. (**b**) The bar graph shows the average current amplitude at different ion concentrations. When comparing the currents of extracellular solutions with different ion concentrations, the control is the 10 mM Na^+^ concentration. Whenever the ion concentration is higher than 100 mM NH_4_^+^, the cell dies, so no higher concentration gradient was tested. All data are expressed as mean ± SEM (*n* = 5 cells).

**Figure 2 molecules-28-06815-f002:**
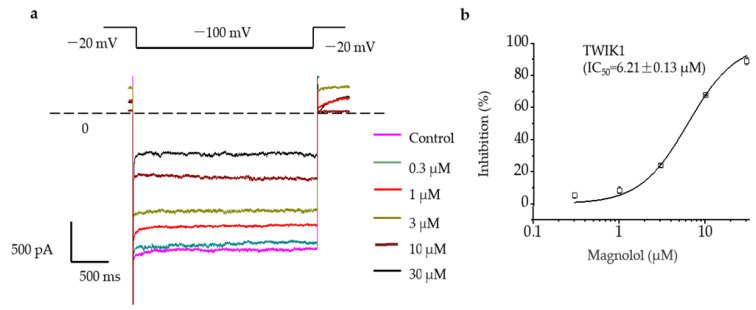
Concentration-dependent inhibition of Magnolol. (**a**) The upper panel indicates the voltage protocol. The lower panel shows examples of current traces of different magnolol concentrations activated by 100 mm NH4^+^. (**b**) Fit of the Hill equation of concentration-dependent inhibition of magnolol to TWIK1. All data are expressed as mean ± SEM (*n* = 5 cells).

**Figure 3 molecules-28-06815-f003:**
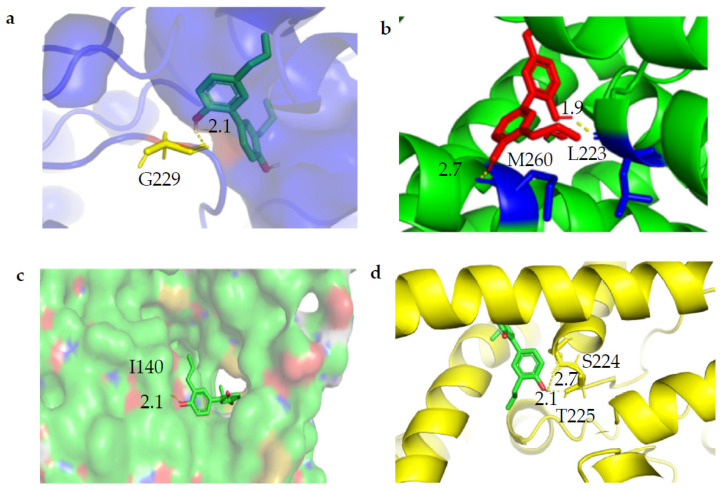
The optimal confirmation was obtained by docking the magnolol molecule with TWIK1. (**a**) Potential hydrogen bonding between magnolol and G229 at the helix of the P2 hole. (**b**) Magnolol forms hydrogen bonds with L223 at the helix of the P2 pore and M260 at the inner helix of M4. (**c**) Potential hydrogen bonding between magnolol and I140 at the transmembrane helix M2. (**d**) Potential hydrogen bonds between magnolol and S224 and T225 at the P2 pore helix. The dotted line in the figure represents hydrogen bonds. The number represents the key length, and the unit is Å.

**Figure 4 molecules-28-06815-f004:**
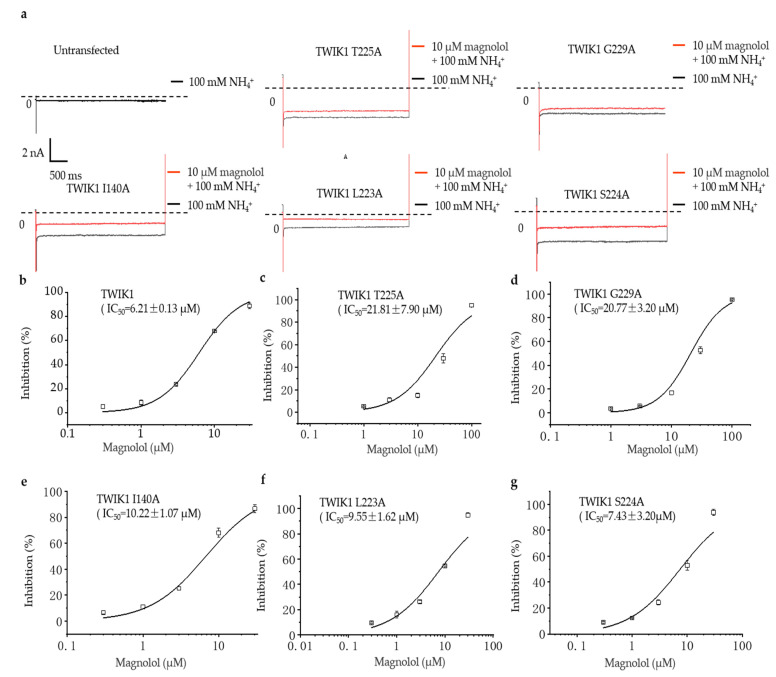
Concentration-dependent inhibitory effect of magnolol against TWIK1 mutations. (**a**) The upper panel shows the example current traces of different TWIK1 mutations at 10 μM magnolol concentrations under the activation of 100 mM NH4^+^. (**b**) Wild-type TWIK1, (**c**) T225A, (**d**) G229A, (**e**) I140A, (**f**) L223A, and (**g**) S224A suggest the IC_50_ of magnolol.

## Data Availability

Not applicable.

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
