# Peer review of "The Inhibitory Effect of Magnolol on the Human TWIK1 Channel Is Related to G229 and T225 Sites"

_molecules, 2023, doi:10.3390/molecules28196815_

Round 1
Reviewer 1 Report
I found this paper very interesting and have only minor comments and suggestions.
Line 19; consider adding spaces before and after the “±”
Line 20; ‘mutant’ > ‘mutants’
Lines 20-21; add the “A” to mutant names, eg., ‘G229’ > ‘G229A’
Line 23; ‘magnolols’ > ‘magnolol’
Line 24; ‘Tow’ > ‘Two’, add ‘channel’ after ‘potassium’, G229 and T225 cannot be key words
Line 46; ‘Tow’ > ‘Two’
Lines 46-47; ’encoded by 15 genes’ is confusing. Please clarify that there are 15 members of the two-pore-domain potassium channels encoded by 15 genes.
Line 72; please explain what ‘Chinese medicine monomer cells’ are. It seems like the authors meant to say that cells were treated with Chinese medicine monomers.
Lines 74-76 and 86; This is a very specific formulation. Could the authors elaborate on why they specifically expected hydrogen bonding to residues like G229 (glycine can only form hydrogen bonds via its backbone atoms) would be important? If specific reasons cannot be given, I would advise reformulating this statement more generally. Furthermore, based on how the introduction was written, it is hard to determine whether the authors always intended to test the effect of magnolol specifically, or if the introduction was written this way after the authors identified magnolol in a screen of many compounds.
Lines 128-129; are the error bars too small to be seen? In that case, please state this, or consider plotting the individual data points.
Line 140; ‘This shows that magnolol interacts’ could be rephrased. please use ‘suggests’ rather than ‘shows’ since the docking results would need to be confirmed experimentally to make this a definitive statement.
Line 155; Figure 3; The text on the figures is hard to read in some cases, please consider moving the numbers around to allow greater contrast with the background.
Line 156; Please be consistent in the use of either one-letter or three-letter abbreviations for the amino acid residues (G229 vs GLY229).
Line 162; ‘mutation’ > ‘mutants of’
Line 167; please use consistent notation for referring to the amino acid residues (here the numbers are superscripts)
Line 197; Figure 4; adding the plot for the wild-type protein to this figure would allow simpler comparison.
Line 199; space after ‘100’ and after ‘mM’
Lines 214-215; Could the authors comment on the relevance of determining inhibition constants under abnormal conditions (i.e., high ammonium concentrations).
Line 219; italicize ‘M. officinalis’, ‘biund’ > ‘binds’
Line 225; ‘combines’ > ‘interacts’
Line 242; ‘it is found’ > ‘it was found’
Line 252; Some additional discussion; The arguments and interpretation of the experimental data seem appropriate. However, despite identifying residues that potentially interact with magnolol (neither docking nor alanine scanning can confirm unambiguously that the mechanism of interaction is as proposed by the authors) it is not clear what this information can be used for. It would be helpful if the authors could explain their vision for how this information can be used further. Since these are human proteins, it seems unlikely that one would want to engineer them, but perhaps this information could be used to interpret inter-individual variability based on known SNPs in the gene encoding TWIK1. A short discussion of this would be highly appreciated.
I do not think the authors necessarily need to make more mutants, but it would be valuable to read a discussion on the limitations of alanine scanning. Sometimes one can learn more from more conservative mutations (e.g., T>S rather than T>A, M>L rather than M>A, etc.). Is anything else known about variation between humans at these positions? By now many millions of genomes have been sequenced and it would be interesting to know whether there are any prominent variations at these positions. Again, I am not suggesting the authors make more mutants for this paper, just that they add a discussion on the limitation of alanine scanning when used in isolation.
Line 264; ‘HEK 293’ should not have a space in it
Line 271; ‘mutation’ > ‘mutated’
Line 271; ‘cDNAs’ confused me. if the cells were transfected with cDNA, please describe its production. From the rest of this paragraph, it seems like the authors mean that the cells were transfected with the mutant plasmids they generated. Please clarify.
Line 277, 278, and 280; ‘mother liquid’ > ‘stock solution’
Line 279; space in ‘0.09 mM’
Line 290; space after ‘base’
Line 295; ‘experiment’ > ‘experiments’
Line 300; I would assume the same mechanism by which the presence of glucose depolarizes TWIK1 in pancreatic b cells is not relevant in HEK293 cells, but it would be appreciated if the authors could comment on this.
I pointed out some parts of the text that were harder to understand and tried to make suggestions for improving them. I apologise for any misunderstandings that may have led to inappropriate comments.
Author Response
We appreciate the reviewer’s enthusiasm for the work.
1.Line 19; consider adding spaces before and after the “±”.
Response 1: Thank you very much. We modified according to suggestions (marked with bright yellow background and red font in Line 19 ).
2.Line 20; ‘mutant’ > ‘mutants’
Response 2: Thank you very much. We modified according to suggestions (marked with bright yellow background and red font).
3.Lines 20-21; add the “A” to mutant names, eg., ‘G229’ > ‘G229A’
Response 3: Thank you very much. We modified according to suggestions (marked with bright yellow background and red font).
4.Line 23; ‘magnolols’ > ‘magnolol’
Response 4: Thank you very much. We modified according to suggestions (marked with bright yellow background and red font).
- Line 24; ‘Tow’ > ‘Two’, add ‘channel’ after ‘potassium’, G229 and T225 cannot be key words
Response 5: Thank you very much. We modified according to suggestions (marked with bright yellow background and red font).
6.Line 46; ‘Tow’ > ‘Two’
Response 6: Thank you very much. We modified according to suggestions (marked with bright yellow background and red font).
7.Lines 46-47; ’encoded by 15 genes’ is confusing. Please clarify that there are 15 members of the two-pore-domain potassium channels encoded by 15 genes.
Response 7: Thank you very much.This sentence has been revised as “Two-pore-domain potassium channel (K2P) is a background potassium channel encoded by 15 KCNK genes. ”
- Line 72; please explain what ‘Chinese medicine monomer cells’ are. It seems like the authors meant to say that cells were treated with Chinese medicine monomers.
Response 8: Thank you. It is a method of whole-cell patch clamp technique to carry out experiments on traditional Chinese medicine monomers. According to your suggestion, we have made changes as “Then, the concentration-dependent inhibition of magnolol on TWIK1 was observed by whole-cell patch clamp experiment. ”
- Lines 74-76 and 86; This is a very specific formulation. Could the authors elaborate on why they specifically expected hydrogen bonding to residues like G229 (glycine can only form hydrogen bonds via its backbone atoms) would be important? If specific reasons cannot be given, I would advise reformulating this statement more generally. Furthermore, based on how the introduction was written, it is hard to determine whether the authors always intended to test the effect of magnolol specifically, or if the introduction was written this way after the authors identified magnolol in a screen of many compounds. Lines 128-129; are the error bars too small to be seen? In that case, please state this, or consider plotting the individual data points.
Response 9: Thank you very much. According to your suggestion, we agreed your more general statement about amino acid residues. We only discuss one compound magnolol in the whole article, so we changed the statement at the end of the introduction as“The mutation was used to investigate whether the effect of magnolol on TWIK1 was related to the binding site between amino acid residues in the selective filter.” .
Lines 128-129,Yes , the error bars too small to be seen. If we magnified the figure, they will be seen.
- Line 140; ‘This shows that magnolol interacts’ could be rephrased. please use ‘suggests’ rather than ‘shows’ since the docking results would need to be confirmed experimentally to make this a definitive statement.
Response 10: Thank you very much. We modified according to suggestions (marked with bright yellow background and red font).
11.Line 155; Figure 3; The text on the figures is hard to read in some cases, please consider moving the numbers around to allow greater contrast with the background.
Response 11: Thank you very much. According to the suggestion, Figure 3 has been modified.
12.Line 156; Please be consistent in the use of either one-letter or three-letter abbreviations for the amino acid residues (G229 vs GLY229).
Response 12: Thank you very much.We modified the full text by using single letter abbreviation for amino acid residues.
- Line 162; ‘mutation’ > ‘mutants of’
Response 13: Thank you very much. According to the suggestion, it has been revised in Line 159.
- Line 167; please use consistent notation for referring to the amino acid residues (here the numbers are superscripts)
Response 14: Thank you very much.The format of amino acid residues in the full text has been unified.
- Line 197; Figure 4; adding the plot for the wild-type protein to this figure would allow simpler comparison.
Response 15: Thank you very much. We have added the wild-type “TWIK1” to Fig. 4b.
16.Line 199; space after ‘100’ and after ‘mM’
Response 16: Thank you very much. it has been revised.
17.Lines 214-215; Could the authors comment on the relevance of determining inhibition constants under abnormal conditions (i.e., high ammonium concentrations).
Response 17: Thank you very much. It is hardly to observe the current of TWIK1 in cultured mammalian cells since the current is very small. In the past, the main method of activation was to increase the current by mutating some amino acid residue. In Lines 66-68 of the introduction, we explain the reasons for activation with high ammonia.That is it was found that the inward NH4+ current of the TWIK1 channel was 40 times larger than the K+ current in a heterologous expression system.
18.Line 219; italicize ‘M. officinalis’, ‘biund’ > ‘binds’
Response 18: Thank you very much. We modified according to suggestions (marked with bright yellow background and red font).
- Line 225; ‘combines’ > ‘interacts’
Response 19: Thank you very much. We modified according to suggestions (marked with bright yellow background and red font).
- Line 242; ‘it is found’ > ‘it was found’
Response 20: Thank you very much. We modified according to suggestions (marked with bright yellow background and red font).
- Line 252; Some additional discussion; The arguments and interpretation of the experimental data seem appropriate. However, despite identifying residues that potentially interact with magnolol (neither docking nor alanine scanning can confirm unambiguously that the mechanism of interaction is as proposed by the authors) it is not clear what this information can be used for. It would be helpful if the authors could explain their vision for how this information can be used further. Since these are human proteins, it seems unlikely that one would want to engineer them, but perhaps this information could be used to interpret inter-individual variability based on known SNPs in the gene encoding TWIK1. A short discussion of this would be highly appreciated.
I do not think the authors necessarily need to make more mutants, but it would be valuable to read a discussion on the limitations of alanine scanning. Sometimes one can learn more from more conservative mutations (e.g., T>S rather than T>A, M>L rather than M>A, etc.). Is anything else known about variation between humans at these positions? By now many millions of genomes have been sequenced and it would be interesting to know whether there are any prominent variations at these positions. Again, I am not suggesting the authors make more mutants for this paper, just that they add a discussion on the limitation of alanine scanning when used in isolation.
Response 21: Thank you very much. According to the suggestion, we revised the last part of the discussion, 247-252 “In this work, the mechanism of magnolol inhibiting TWIK1 channel was studied. Furthermore, T225 and G229 residues were identified by site mutation as important amino acid residues that affected magnolol's inhibition of TWIK1 channel. This provides a research direction for us to discuss whether magnolol can affect the treatment of diabetes through specific amino acid residues, and also provides a new way for screening candidate drugs targeting TWIK1 channel.” .
22.Line 264; ‘HEK 293’ should not have a space in it
Response 22: Thank you very much.We modified according to suggestions (marked with bright yellow background and red font).
23.Line 271; ‘mutation’ > ‘mutated’
Response 23: Thank you very much. We modified according to suggestions (marked with bright yellow background and red font).
- Line 271; ‘cDNAs’ confused me. if the cells were transfected with cDNA, please describe its production. From the rest of this paragraph, it seems like the authors mean that the cells were transfected with the mutant plasmids they generated. Please clarify.
Response 24: Thank you very much. This part has been modified, in Lines 259-266“TWIK1 stable cell line and Human Embryonic Kidney (HEK293) cells were cultured in a solution containing 10% (v/v) fetal bovine serum (FBS), 1% (v/v) P/S (100U Penicillin and 0.1 mg/mL streptomycin). Additional 100 μg/mL G418 were added to culture TWIK1 stable cell line. Cells were passaged and transiently transfected with 1 µg WT and T225A, G229A, L223A, S224A, and I140A mutations using transient transfection reagents (Attractene Transfection Reagent, Qiagen, Hilden, Germany), while CD8 plasmid was co-transfected. The successfully transfected cells were identified by labeling with CD8-specific antibody-coated microspheres (Dynal, Oslo, Norway).”.
25.Line 277, 278, and 280; ‘mother liquid’ > ‘stock solution’
Response 25: Thank you very much. We modified according to suggestions (marked with bright yellow background and red font).
26.Line 279; space in ‘0.09 mM’
Response 26: Thank you very much. We modified according to suggestions (marked with bright yellow background and red font).
27.Line 290; space after ‘base’
Response 27: Thank you very much. We modified according to suggestions (marked with bright yellow background and red font).
- Line 295; ‘experiment’ > ‘experiments’
Response 28: Thank you very much. We modified according to suggestions (marked with bright yellow background and red font).
29.Line 300; I would assume the same mechanism by which the presence of glucose depolarizes TWIK1 in pancreatic b cells is not relevant in HEK293 cells, but it would be appreciated if the authors could comment on this.
Response 29: Thank you very much. We perfused TWIK1 cells with high concentration of glucose, and our experiments showed that high concentration of glucose also inhibited TWIK1 channels.
Reviewer 2 Report
In summary, the study presents relevant information about the mechanism of TWIK1 channel inhibition by magnolol, highlighting the crucial magnolol binding sites in the channel. However, some improvements in terms of contextualization and broader discussion could further improve the work. Furthermore, it is critical to explore the clinical implications of these findings and to consolidate the scientific validity of the study.

Author Response
We appreciate the reviewer’s enthusiasm for the work.
- I suggest introducing spaces between symbols, for example: IC50 = 6.21 ± 0.13 μM) - review all text.
Response 1: Thank you very much. The whole text has been revised.(marked with bright yellow background and red font).
- Some words should be revised, for example: line 23 – magnolols is magnolol; -review all text.
Response 2: Thank you very much. We modified according to suggestions (marked with bright yellow background and red font).
- Abstract: I suggest a final sentence that represents the conclusion of the study.
Response 3: Thank you very much. We have changed the last sentence of our abstract summarizes our research conclusion,that is “Thus, we conclude that the inhibition of the TWIK1 channel by magnolol is related to G229 and T225 on the P2- pore helix.”.And we have added the conclusion in Line 253-254 in the discussion section.
4.Introduction: I liked the introduction, it is objective and addresses the necessary content.
Response 4: Thank you very much. Thank you for your review.
5.What are 'Chinese medicine monomer cells'? explain.
Response 5: Thank you. It is a method of whole-cell patch clamp technique to carry out experiments on traditional Chinese medicine monomers. According to your suggestion, we have made changes as “Then, the concentration-dependent inhibition of magnolol on TWIK1 was observed by whole-cell patch clamp experiment. ”.
6.Results: Include error bars in all figures that do not have them. Improve the texts present in the figures - they are too small. A suggestion would be to put them in a Note.
Response 6: Thank you very much. According to your suggestion, we made the error line thicker, but did not make the mean value box bigger, because it would cover the error line. So we set the mean value box from black solid to hollow.
7.All statements regarding molecular coupling must be referred to as "suggest", "appear"... as it still requires experimental confirmation before a more conclusive statement can be made.
Response 7: Thank you very much. The whole text has been revised.
8.I missed a referral in the discussion that refers to the application in medicine. I suggest including how the results of the study can contribute to the area.
Response 8: Thank you very much. We have added the application in medicine and the results of the study contribution to the area in Line 247-252 in the discussion section”In this work, the mechanism of magnolol inhibiting TWIK1 channel was studied. Furthermore, T225 and G229 residues were identified by site mutation as important amino acid residues that affected magnolol's inhibition of TWIK1 channel. This provides a research direction for us to discuss whether magnolol can affect the treatment of diabetes through specific amino acid residues, and also provides a new way for screening candidate drugs targeting TWIK1 channel.”